

# Identification and differential expression of piRNAs in the gonads of Amur sturgeon (*Acipenser schrenckii*)

Lihong Yuan[1,2], Linmiao Li[2], Xiujuan Zhang[2], Haiying Jiang[2] and Jinping Chen[2]

[1] School of Life Sciences and Biopharmaceutics, Guangdong Pharmaceutical University, Guangzhou, China
[2] Guangdong Key Laboratory of Animal Conservation and Resource Utilization, Guangdong Public Laboratory of Wild Animal Conservation and Utilization, Guangdong Institute of Applied Biological Resources, Guangzhou, China

Corresponding author
Jinping Chen, chenjp@giabr.gd.cn

## ABSTRACT

**Objective**. Sturgeons are considered living fossils, and have a very high conservation and economic value. Studies on the molecular mechanism of sturgeon gonadal development and sex differentiation would not only aid in understanding vertebrate sex determination but also benefit sturgeon aquaculture. Piwi-interacting RNAs (piRNAs) have been shown to function in germline or gonadal development. In this study, we performed small RNA deep sequencing and microarray hybridization to identify potential sturgeon piRNAs.

**Methods**. Male and female sturgeon gonads were collected and used for small RNA sequencing on an Illumina HiSeq platform with the validation of piRNA expression by microarray chip. The program Bowtie and $k$-mer scheme were performed to filter small RNA reads and discover potential sturgeon piRNAs. A known piRNA database, the coding sequence (CDS), 5′ and 3′ untranslated region (UTR) database of the *A. Schrenckii* transcriptome, Gene Ontology (GO) database and KEGG pathway database were searched subsequently to analyze the potential bio-function of sturgeon piRNAs.

**Results**. A total of 875,679 putative sturgeon piRNAs were obtained, including 93 homologous to known piRNAs and hundreds showing sex-specific and sex-biased expression. Further analysis showed that they are predominant in both the ovaries and testes and those with a sex-specific expression pattern are nearly equally distribution between sexes. This may imply a relevant role in sturgeon gonadal development. KEGG pathway and GO annotation analyses indicated that they may be related to sturgeon reproductive processes.

**Conclusion**. Our study provides the first insights into the gonadal piRNAs in a sturgeon species and should serve as a useful resource for further elucidation of the gene regulation involved in the sex differentiation of vertebrates. These results should also facilitate the technological development of early sex identification in sturgeon aquaculture.

## INTRODUCTION

Sturgeons (order: Acipenseriformes, infraclass: Chondrostei) are referred to as living fossils and have considerable value in aquaculture as sturgeon eggs (caviar) (*Bemis, Findeis & Grande, 1997*; *Ludwig, 2008*). Sturgeons contain 25 caviar-producing species, 17 of which are members of the *Acipenser* genus (*Bemis, Findeis & Grande, 1997*). Due to the huge profits associated with the sale of sturgeon caviar, over-exploitation of wild stocks occurred worldwide throughout the 20th century, and major sturgeon fisheries are in decline (*Pikitch et al., 2005*). Currently, all *Acipenser* species appear on the IUCN Red List of Critically Endangered Species (*Ruban & Qiwei, 2013*) and under Appendices I or II of CITES (Convention on International Trade in Endangered Species) (*Dongol, 2011*). Given the decline of wild stocks and conservation, great efforts have been made to develop commercial sturgeon aquaculture to meet the demand for caviar, originally in Europe and North America but more recently in Russia, Iran and China (*Raymakers & Hoover, 2010*; *Wei et al., 2011*). Currently, it is estimated that approximately 50% of the caviar in trade is from farmed stocks (*Bronzi, Rosenthal & Gessner, 2011*).

The rapid development of sturgeon aquaculture has greatly decreased the pressure on wild stocks; however, aquaculture brings new challenges. Two of the primary challenges are performing early sex identification and culling individuals with gonad dysplasia. Due to the long culture period in sturgeon (with an estimated 15 years/generation), approximately 50% of the offspring are profitless males, and up to 30% of offspring have gonad dysplasia (*Krykhtin & Svirskii, 1997*). Performing early sex identification, increasing the proportion of female sturgeon and decreasing gonad dysplasia may significantly improve sturgeon farming profits. Currently, the methods used in sturgeon sex and reproductive stage determination (i.e., laparoscopy, ultrasonography, histology and sex steroid analyses *Falahatkar et al., 2011*; *Petochi et al., 2011*) are highly dependent on technician experience and are restricted by age and gonad maturity (*Devlin & Nagahama, 2002*; *Masoudifard et al., 2011*). Therefore, a better understanding of the processes that regulate sexual development, especially gonadogenesis and gametogenesis, may provide novel targets in sturgeon aquaculture. Furthermore, in contrast to mammals, the sex of lower bony fish is unstable and may be affected by many factors such as the environment and hormones (*Devlin & Nagahama, 2002*). However, due to the complexity of the number and ploidy of their chromosomes and lack of knowledge regarding sex chromosome differentiation, the study of sex determination and differentiation in sturgeons at the molecular level is still difficult.

Piwi-interacting RNAs (piRNAs), a distinct class of 26–32 nt non-coding RNAs, have been shown to function in germline development, transposon silencing and epigenetic regulation mediated by Piwi proteins (*Ashe et al., 2012*; *Houwing, Berezikov & Ketting, 2008*; *Juliano, Wang & Lin, 2011*; *Malone & Hannon, 2009*; *Ross, Weiner & Lin, 2014*; *Vagin et al., 2006*). Recently the piRNA biology has been expanded rapidly in not only embryonic patterning, germ cell specification, but also in stem cell biology, neuronal activity and metabolism (*Rojas-Ríos & Simonelig, 2018*). Studies have shown that piRNAs are expressed in both the male and female germlines of *Caenorhabditis elegans*, *Drosophila melanogaster*,

*Danio rerio* and *Xenopus laevis* and in the male germlines of mammals and birds but have limited expression in the early female germlines of mammals (*Ashe et al., 2012*; *Ha et al., 2014*; *Houwing, Berezikov & Ketting, 2008*; *Juliano, Wang & Lin, 2011*; *Lau et al., 2009*; *Li et al., 2013*; *Yang et al., 2013*; *Roovers et al., 2015*; *Wilczynska et al., 2009*; *Williams et al., 2015*). A deficiency in the genes required for piRNA biogenesis affects the regulation of gene silencing, cell differentiation and gonadal development in animals. However, the distribution of piRNAs in sturgeon gonads and the function in sex differentiation remains unclear.

*Acipenser schrenckii* (Amur sturgeon) is an economically important sturgeon species in China, and the wild stocks are mainly distributed in the Amur River, Songhua River and Heilong River (*Li, Liu & Xie, 2012*). Sturgeon small RNA transcriptome and gene expression patterns in many tissues (including the gonads) have been assayed using RNA-Seq technology, and a batch of sex-biased RNAs have been identified (*Jin et al., 2015*; *Yuan et al., 2014*; *Zhang et al., 2016a*; *Zhang et al., 2016b*). These studies support sturgeon gonad development research. In the present study, we first analyzed the putative piRNAs of Amur sturgeon gonads using the Illumina sequencing platform. We combined these data with the results of microarray expression validation to identify the putative sturgeon-specific and/or sex-specific piRNAs and illustrate the potential role of putative piRNAs on sturgeon gonadal development and sex differentiation. This study should provide information regarding piRNAs in gonads, help to reveal the mechanism of the sex duality of sturgeon, and contribute to identifying gender-related bio-markers for use in sturgeon aquaculture.

## MATERIALS AND METHODS

### Ethics statement

The protocol was approved by the Committee on the Ethics of Animal Experiments of Guangdong Institute of Applied Biological Resources (GIABR2014008). Individual sturgeon were immersed in water with 10-4 (v/v) eugenol for approximately 1–3 min for euthanasia, according to the AVMA guidelines (*Leary et al., 2013*). All efforts were made to minimize suffering.

### Sample and RNA preparation

In this study, we used 3-year-old Amur sturgeons (*Acipenser schrenckii*), whose sex could be identified accurately by laparoscopy and histology. The animals were obtained from the Engineering and Technology Center of Sturgeon Breeding and Cultivation of the Chinese Academy of Fishery Science (Beijing, China). The testes and ovaries of six 3-year-old Amur sturgeons (three males and three females) were collected. Total RNA was extracted from tissue samples separately with RNAiso reagent (TaKaRa, Shiga, Japan) according to the manufacturer's instructions. The RNA concentrations were measured using a Qubit RNA Assay Kit in Qubit 2.0 Fluorometer (Life Technologies, Carslbad, CA, USA), and RNA purity was assessed using a Nano Photometer spectrophotometer (IMPLEN, Westlake Village, CA, USA). RNA integrity was inspected using an RNA Nano 6000 Assay Kit and Bioanalyzer 2100 system (Agilent Technologies, Santa Clara, CA, USA).

## Small RNA library preparation and sequencing

Four RNA samples (two testes and two ovaries, 3 μg RNA of each) were used for the construction and sequencing of small RNA libraries. In brief, NEB 3′ SR Adaptor ligation, SR RT Primer hybridization and NEB 5′ SR Adaptor ligation were performed according to the NEBNext Multiplex Small RNA Sample Preparation Set (Illumina, San Diego, CA, USA) protocol. After first strand cDNA synthesis using M-MuLVA Reverse Transcriptase (RNase H⁻) by PCR, the 140 bp–160 bp products (with adaptors on both sides) were separated on an 8% PAGE gel and quantified using an Agilent Bioanalyzer 2100 system. Then, a cluster of index-coded samples was generated using a TruSeq SR Cluster Kit v3-cBot-HS (Illumina, San Diego, CA, USA) and sequenced on an Illumina HiSeq 2000 platform. Finally, 50 bp single-end reads were generated.

## Small RNA annotation and piRNA identification

In this study, the testis and ovary transcriptomes of *A. Schrenckii* (*Jin et al., 2015*) were used as reference sequences for small RNA annotation. After removal of unclean reads (adapters, low quality reads, reads containing '*n*', and redundant reads), clean unique reads were mapped onto the *A. schrenckii* transcriptome reference sequences using the program Bowtie (*Langmead et al., 2009*) with no mismatches allowed. Perfectly mapped reads were filtered by three successive steps to remove small RNA elements: (1) searched against Sanger miRBase (Release 19) to exclude conserved miRNAs; (2) screened against Rfam (http://rfam.sanger.ac.uk/) and RepeatMasker (http://www.repeatmasker.org/) with Bowtie to filter the sequences originating from rRNA, tRNA, snRNA, snoRNA and repeats; (3) analyzed by miREvo, mirdeep2 and MirCheck to remove the potential novel miRNA reads. The detailed process of small RNA filter was described in our previous study (*Yuan et al., 2014*).

The remaining reads with lengths of 26–32 nt were used for piRNA discovery. A *k*-mer scheme relied on the training sets of non-piRNA and the piRNA sequences of five model species (rat, mouse, human, fruit fly and nematode), was applied as previously described (*Zhang, Wang & Kang, 2011*). Then, it was compared with the existing 'static' scheme on the basis of the position-specific base usage. Putative novel piRNAs were scanned against piRNABank, (http://pirnabank.ibab.ac.in) using Bowtie with no mismatches allowed to identify the orthologs of known piRNAs. The relative frequencies of piRNA nucleotide utility reads were analyzed. Subsequently, the putative piRNAs were functionally annotated using the coding sequence (CDS), 5′ and 3′ untranslated region (UTR) database of the *A. Schrenckii* transcriptome with TransDecoder software (https://github.com/trinityrnaseq/trinityrnaseq/wiki/Coding-Region-Identification-in-Trinity-Assemblies).

## piRNA microarray and data analysis

To validate the expression of putative piRNAs identified by Illumina sequencing, we selected: (1) 1,092 that exhibited significantly different expression in ovaries vs. testes with |logFC| ≥ 1.5, $P \leq 0.01$, Padj ≤ 0.05, and read counts >10; and (2) 779 highly expressed putative piRNAs with read counts 50∼4,646 in ovaries or 50∼8,177 in testes. For each of

the six RNA samples (three ovaries and three testes, extracted above) assessed, 4 μg of total RNA were used to hybridize with the microarray chip.

A piRNA microarray was manufactured by RIBOBIO (Guangdong, China), and each piRNA probe had three replicates. The chip was hybridized with single-color labeling (Cy5) RNAs according to the manufacturer's protocol, with no modifications. Microarray results were extracted using a laser scanner (GenePix 4000B, Molecular Device) and digitized using Array-Pro image analysis software (Media Cybernetics, Rockville, MD, USA). Raw data were subtracted using the background matrix, and spots with CV [(standard deviation)/(signal intensity)] < 0.3 were normalized using a quantile normalization method to remove system related variations, including sample amount variations and signal gain differences of the scanners and to faithfully reveal the biological variations (*Bullard et al., 2010*). The medians of repeated data (normalized intensity) were used for statistical analyses with One-Way ANOVA method, and the expression of piRNAs was deemed significant with the criterion $|logFC| \geq 1$ and $P \leq 0.05$.

### piRNA-generating gene prediction and annotation

Putative piRNAs were mapped onto the *A. schrenckii* transcriptome using the program Bowtie with no mismatches allowed to identify piRNA-generating genes. The piRNA-generating genes were mapped to Gene Ontology (GO) database (http://www.geneontology.org/) by Interproscan (*Zdobnov & Apweiler, 2001*) and the KEGG pathway database (http://www.genome.jp/kegg/) by BLASTX at *E* values 1e−10 (*Kanehisa et al., 2006*), respectively by two levels, (1) total of piRNA-generating genes and (2) sex-specific expression (only expressed in testes or ovaries and with a read count ≥ 10) or sex-biased expression of piRNA-generating genes in ovaries and testes ($|logFC| \geq 1$ and $P \leq 0.05$ obtained by microarray). Finally, the enriched functional groups or pathways were obtained with corrected $P < 0.05$.

## RESULTS

### Sequencing and statistics of small RNA reads

A total of $7.7 \times 10^6$–$8.6 \times 10^6$ reads were sequenced from four small RNA libraries, with an error rate of 0.01%, Q30 > 97.3% and GC content of ∼48% (Table S1). Then, approximately $7.3 \times 10^6$–$8.1 \times 10^6$ high-quality small RNA reads (>94% in each library) were obtained after removal of ambiguous reads (Table S1). The size distribution and frequency percentage of small RNA reads are shown in Fig. S1 and, of these, the potential piRNA reads (26–30 nt) were the major component (approximately 58%). About $3.1$–$3.4 \times 10^6$ small RNA reads mapped to the *A. schrenckii* transcriptome reference sequences (*Jin et al., 2015*) were with a perfect match (Table S1). A total of $3.5$–$6.7 \times 10^4$ reads were mapped to at least one putative Amur sturgeon miRNA precursor by searching against miRBase, corresponding to 1.0–2.2% perfectly matched small RNA reads (Table 1). Subsequent small RNA filtering indicated that other non-coding RNAs (rRNA, tRNA, snRNA and snoRNA) and repeat sequences were approximately 1.0–1.5% and 0.3–0.4%, respectively. Then, $5.3$–$6.6 \times 10^3$ (0.2%) potential novel miRNA reads specific to *A. schrenckii* were detected with miREvo (Table 1).

Yuan et al. (2019), *PeerJ*, DOI 10.7717/peerj.6709

**Table 1** Small RNA annotation of *Acipenser schrenckii*.

| Sample description | Mapped sRNA | Conserved_ miRNA | rRNA | tRNA | snRNA | snoRNA | Repeats | Novel_ miRNA | piRNA | Uniq piRNA | Other |
|---|---|---|---|---|---|---|---|---|---|---|---|
| Ovary 1 | 3,367,185 (100.00%) | 35,186 (1.04%) | 25,313 (0.75%) | 1 (0.00%) | 21,004 (0.62%) | 862 (0.03%) | 10,175 (0.30%) | 6,055 (0.18%) | 566,402 (16.75%) | 272,963 (8.1%) | 2,704,564 (80.32%) |
| Ovary 2 | 3,303,306 (100.00%) | 52,948 (1.60%) | 28,912 (0.88%) | 0 (0.00%) | 3,595 (0.11%) | 826 (0.03%) | 10,615 (0.32%) | 6,401 (0.19%) | 595,130 (18.02%) | 279,287 (8.45%) | 2,604,879 (78.86%) |
| Testis 1 | 3,164,300 (100.00%) | 39,703 (1.25%) | 44,586 (1.41%) | 1 (0.00%) | 4,478 (0.14%) | 641 (0.02%) | 10,560 (0.33%) | 6,552 (0.21%) | 613,914 (19.40%) | 271,914 (8.59%) | 2,443,865 (77.23%) |
| Testis 2 | 3,126,400 (100.00%) | 67,386 (2.16%) | 39,749 (1.27%) | 0 (0.00%) | 6,539 (0.21%) | 578 (0.02%) | 11,264 (0.36%) | 5,340 (0.17%) | 620,324 (19.84%) | 287,659 (9.2%) | 2,375,220 (75.97%) |

## piRNA discovery

According to a *k*-mer scheme analysis which relied on the training sets of non-piRNA and the piRNA sequences of five model species (*Zhang, Wang & Kang, 2011*), we identified total of $5.7–6.2 \times 10^5$ piRNA reads in our sequencing results, that account for 16.6–19.8% of perfectly matched small RNA reads (Table 1). By following steps described in the methods section, 875,679 putative piRNAs from the testis and ovary libraries were predicted, including 93 piRNAs that were homologous to known piRNAs (four in fruit fly, 45 in zebra fish, 34 in human, four in mouse and six in rat, see Table S2). Sturgeon putative piRNAs have a strong U bias at the extreme 5′ position and are enriched for https://www.sciencedirect.com/topics/biochemistry-genetics-and-molecular-biology/adenine (A) at position 10 (Fig. 1A). Further analysis indicated that their rate of sex-specific expression (only expressed in ovary or testis) was up to 87% (767,805/875,679), and the ratio of ovary- vs. testis-specific putative piRNAs was nearly 1:1 (43.99%,385,222/875,679 vs. 43.69%, 382,583/875,679, see Fig. 1B). Moreover, we found a similar percentage of sense and antisense strand putative piRNAs (42.99%, 376,487/875,679 vs. 57.01%, 499,192/875,679, respectively). In total, approximately 42% (175,648 in ovaries and 195,552 in testes) of all putative piRNAs were derived from well-annotated gene regions, 58% were from unannotated genomic regions (Fig. 1C). Of these, only 31,949 (3.65%) and 73,802 (8.43%) matched the 5′ UTR and 3′ UTR, respectively. Abundant piRNAs in gene regions were found in CDS, which represented 265,499 (30.32%) putative piRNAs.

## Sturgeon putative piRNAs are present in both male and female gonads

Small RNA reads of both male and female sturgeon gonads displayed a peak in reads of 26–30 nt in length (55.8% in testes and 56.5% in ovaries, shown in Fig. 2A). This is an expected length distribution for potential novel piRNA reads. Whereas Small RNA reads in five other somatic/non-germline tissues (pool of brain, heart, muscle, liver and spleen tissues of *A. schrenckii* (*Yuan et al., 2014*)) showed a peak in reads of 20–24 nt in length which is an expected length distribution for miRNA reads (∼52.9.4%). Moreover, both ovary and testis samples had a strong 5′ U bias in reads of 20–24 nt (miRNA reads) and 26–30 nt (piRNAs reads) in length, whereas only reads of 20–24 nt among somatic samples had a 5′ U bias (Figs. 2B–2D). The library size of testis and ovary is nearly equal (the ratio of total read counts ≈1:1, Table S1), thus we further compared piRNAs vs miRNAs in sturgeon testis and ovary samples, and obtained a similar ratio of 4.7:1∼5.5:1 (Table 1). This clearly indicates putative piRNAs are far more abundant class of small RNAs in both testis and ovaries compared to miRNAs. Moreover, both piRNA (1:1) and miRNA (1.2:1) showed an equal ratio in testes vs. ovaries (Table 1).

## piRNA-generating genes

A total of 875,679 putative piRNAs were mapped onto the *A. schrenckii* transcriptome (including 122,381 unigenes in testis and 114,527 in ovaries, *Jin et al., 2015*). Total 49,390 piRNA-generating genes were obtained (23,079 in ovaries, 26,311 in testes, and 9634 in both). The mean length of piRNA-generated genes was 967 nt (median 506), with a range

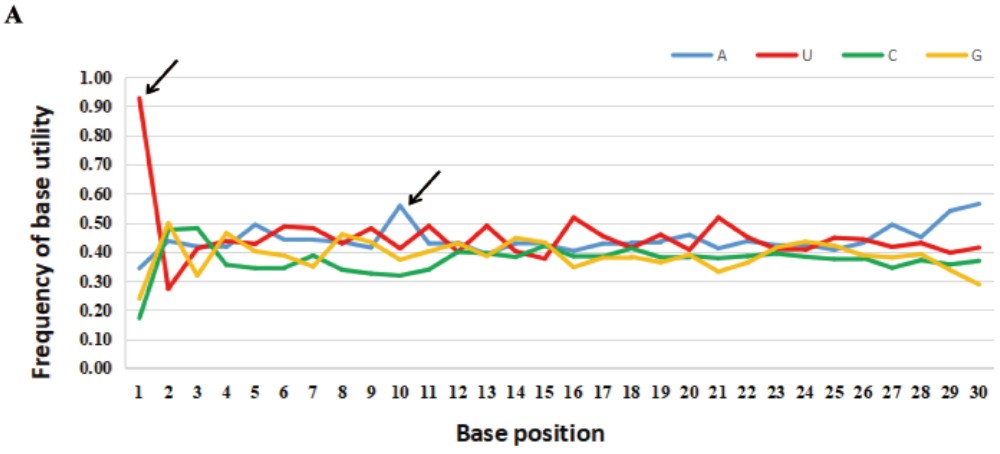

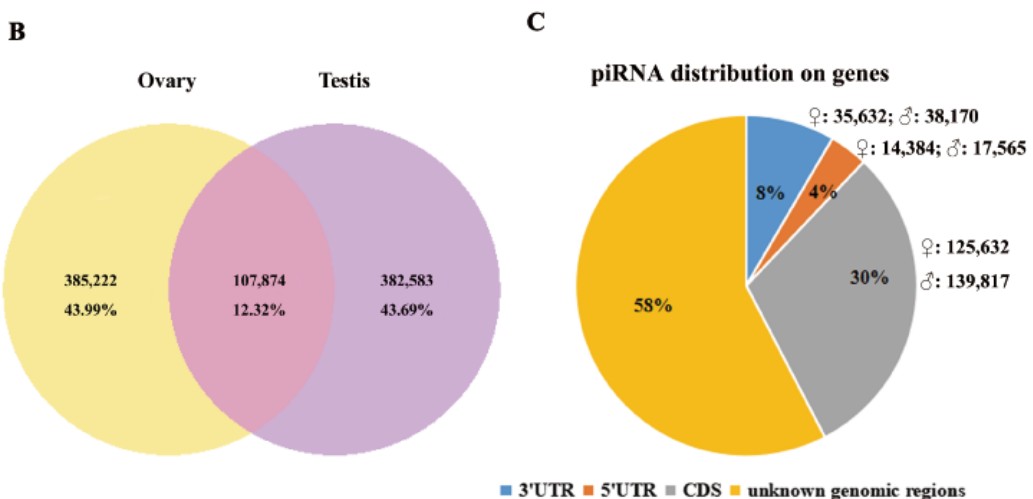

**Figure 1** **Summary of 875,679 sturgeon putative piRNAs.** (A) Base utility analysis of putative piRNAs. Black arrows indicate the preference of 5′U and an adenine at position 10th. (B) Graphical representation of putative piRNAs between ovary and testis libraries. (C) Pie chart summary of putative piRNA distribution.

of 150–16,256 nt. There was a mean of 18 unique piRNAs reads per gene (scale of 1 up to 4,181 piRNAs per gene).

To illustrate the functions of the sturgeon putative piRNAs, gene function of the piRNA-generated genes was annotated by searching against GO database with Interproscan and KEGG pathway database by BLASTX at $E$ values 1e−10. The GO analysis showed 264 enriched GO terms for biological process, molecular function and cellular component, in them, 13 GO terms related to reproduction (such as sexual reproduction, mating/reproductive behavior, post-mating regulation of female receptivity) were obtained (Table S3). Moreover, 29 enriched KEGG pathways, including six involved in aquaculture, such as pathogen infection, multiple diseases, and antigen processing, were identified (Table S3). We further investigated the function of piRNA-generated genes which with

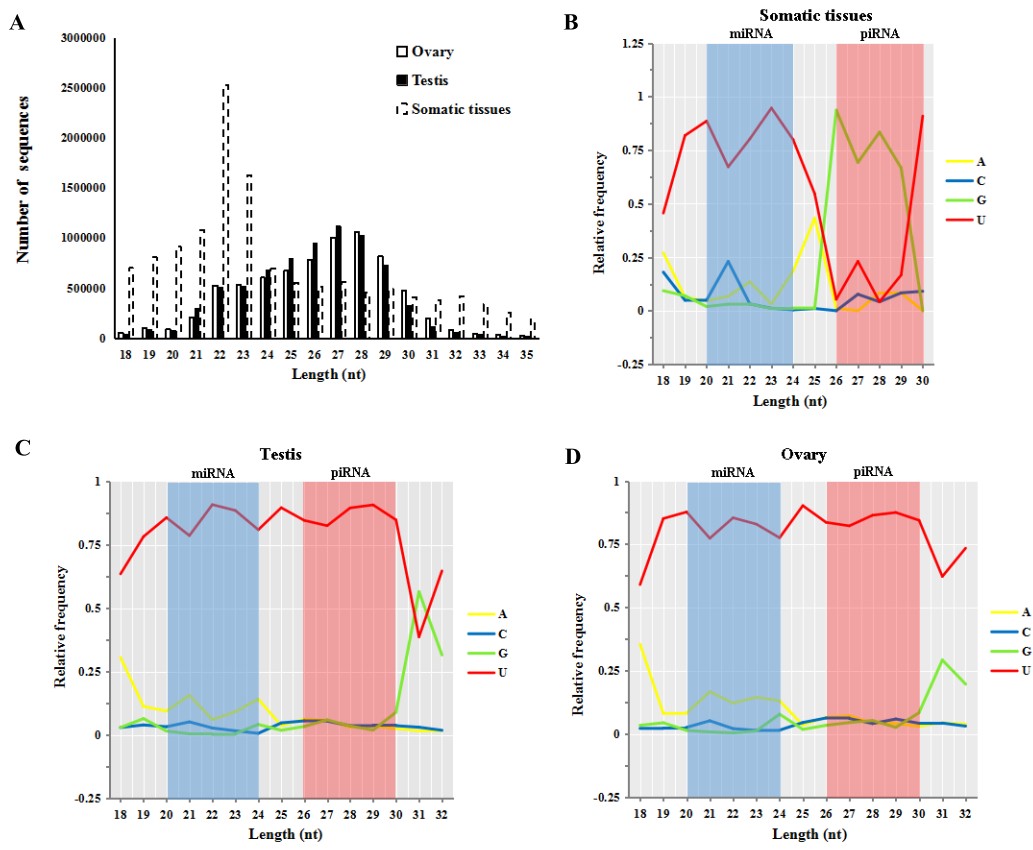

**Figure 2  Base bias analysis of *A. schrenckii* small RNAs.** (A) Length distribution of small RNAs reads from somatic tissues, testes and ovaries of *Acipenser schrenckii*. (B–D) Analysis of the 5′ position nucleotide utility of putative miRNA reads and piRNA reads in somatic tissues, testes and ovaries of *A. schrenckii*. Somatic tissues: data from our previous small RNA transcriptome from a pool of five tissues (brain, heart, muscle, liver and spleen) of *A. schrenckii* (*Yuan et al., 2014*) was used as a control.

sex-specific piRNAs located (1,078 only detected in testis and 1,237 only detected in ovary, with read counts ≥ 10), and more GO terms related to reproductive processes were identified (Fig. 3 and Table S4).

## Validation of putative piRNA expression by microarray

We used an independent microarray platform to validate the expression of 1,871 putative piRNAs (1,092 differently expressed and 779 highly expressed piRNAs, denoted ASY-piRNA-X, where Xs are numerals) identified by Illumina sequencing. A total of 311 showed significantly biased expression between sexes (deemed sex-biased expression) and were clustered into ten clades including five clades up-regulated in ovaries and five clades up-regulated in testes (Fig. S2 and Table S5). Of these, 124 were specifically up-regulated in ovaries with $1 < |logFC| < 8.5$ and $P \leq 0.05$, and 187 were specifically up-regulated in testes with $1 < |logFC| < 4.1$ and $P \leq 0.05$ (Table S5). In sturgeon testes, the putative piRNA with the highest expression level was ASY-piR-1255 with median of normalized intensity 4.04 in testes vs. 0.72 in ovaries (Table S5). In contrast, the most expressed in

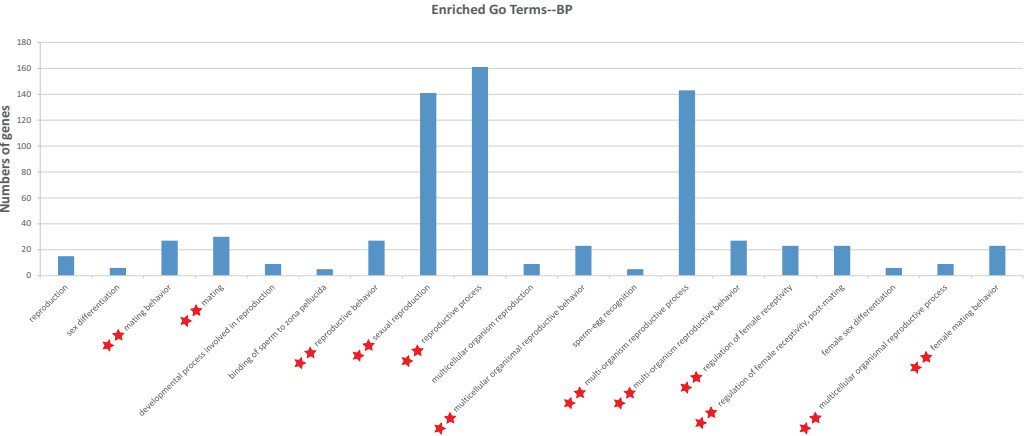

**Figure 3** **Gene ontology (GO) classification annotated for piRNA-generating genes related to reproduction.** \*\*, Enriched GO terms related to reproduction in both levels of sex-specific expressed putative piRNAs and total putative piRNAs. Details about enriched GO terms were listed in Table S3 (at total putative piRNAs level) and Table S4 (at sex-specific putative piRNAs level).

the ovaries was ASY-piR-122 with median of normalized intensity 13.33 in ovaries vs. 4.91 in testes (Table S5). Moreover, 1,335 were co-expressed in both the testes and ovaries (median >1 in both and $P > 0.05$). Further analysis indicated no significant difference in size distribution among sex-biased piRNAs (average 27.6 nt in testes and 27.8 nt in ovaries) and co-expressed piRNAs in both gonads (average 27.5 nt). The Illumina sequencing and microarray expression of putative piRNAs was especially correlated for 311 sex-biased piRNAs ($r = 0.489$, $P = 0.000$). GO and KEGG annotation indicated that the 311 sex-biased expressed piRNAs mainly participate in the metabolic processes of sturgeon gonads.

## DISCUSSION

Here, we report the first identification of putative piRNAs and their expression in male and female sturgeon (*Acipenser schrenckii*) gonads, thus greatly increasing the knowledge of piRNAs in vertebrate gonadal development and/or sex differentiation. We obtained a total of 875,679 putative piRNAs from sturgeon gonads by Illumina sequencing, and hundreds of them showed sex-specific (only expressed in sturgeon testis or ovary) and sex-biased expressed validated by microarray. In addition to basic physiological processes, many reproductive processes (such as sexual reproduction, mating/reproductive behavior, post-mating regulation of female receptivity) as well as pathogen infection and antigen processing are shown associated with the expression of sturgeon putative piRNAs.

The development of an ultrahigh-throughput sequencing technique (RNA-seq) has allowed researchers to discover and analyze the piRNAs of many organisms, and tens of thousands of unique sequences have been identified (*Brennecke et al., 2007*; *Castellano et al., 2015*; *Houwing et al., 2007*; *Williams et al., 2015*). In this study, we obtained a total of 7.3 $\times 10^6$–8.1 $\times 10^6$ high-quality small RNA reads, 58% of which were potential piRNA reads of 26–30 nt in length (Table S1 and Fig. S1). After mapping the reads to the *A. schrenckii*

transcriptome reference sequences, approximately 1.2–2.3% reads and 16.6–19.8% were identified as potential miRNAs and piRNAs, respectively, whereas >75% of the samples were unannotated genomic regions (Table 1). By gathering additional sturgeon genomic and transcriptomic data and comparing against the known sturgeon datasets, the accuracy of sturgeon gene annotation will increase, and much more detailed information on *A. schrenckii* small RNAs can be uncovered from the transcriptome datasets obtained in the present study.

Our study also revealed a large number of sturgeon-specific piRNAs and provided candidates for further study of sturgeon gonadal development. A total of 875,679 sturgeon putative piRNAs were identified, including 93 homologous to known piRNAs. The number of sturgeon putative piRNAs were 10-fold and 40-fold higher than that in zebrafish ovaries and testes (*Houwing et al., 2007*) and normal human testes (*Yang et al., 2013*), respectively. The large number of putative piRNAs obtained in this study was probably due to chromosome ploidy resulting from multiple and independent duplication events (*Fontana et al., 2008*; *Ludwig et al., 2001*). Approximately 86% of reads were sequenced only once, which may have been a result of the absence of genomic information and the incomplete transcriptome annotation for sturgeons.

Further analyses showed that the preference of 5′ U and an https://www.sciencedirect.com/topics/biochemistry-genetics-and-molecular-biology/adenine at position 10 (10A bias) in sturgeon gonads is consistent with a known piRNA biogenesis by Piwi-mediated cleavage (*Brennecke et al., 2007*) (Fig. 1A). Moreover, Small RNA reads with characteristics typical to piRNAs were found in both male and female sturgeon gonads, but not in somatic tissues (Fig. 2A). We show that both expression of sex-specific and distribution on gene regions of putative piRNAs are equally between testis and ovary (Figs. 1B and 1C). The prevailing expression of putative piRNAs in gonads has also been found in many other organisms (*Ashe et al., 2012*; *Houwing et al., 2007*; *Juliano, Wang & Lin, 2011*; *Lau et al., 2009*; *Wilczynska et al., 2009*; *Williams et al., 2015*) and provides insight into the possible roles of sturgeon piRNAs in gametogenesis between sexes. Similar to the zebrafish, sturgeon putative piRNAs and miRNAs have equivalent expression in the testes and ovaries (Table 1), indicating the important role of piRNAs and miRNAs in the development of sturgeon gonads (*Houwing et al., 2007*). Putative piRNAs exhibit significantly higher expression than putative miRNAs in both male and female sturgeon gonads (Table 1), as has also been observed in both male and female zebrafish gonads, in normal adult human testes and mouse testes (*Aravin et al., 2006*; *Beyret, Liu & Lin, 2012*; *Girard et al., 2006*; *Houwing et al., 2007*; *Yang et al., 2013*). In addition, over 87% of piRNAs showed sex-specific expression; in contrast, only approximately 30% of putative miRNAs (223/730) showed sex-specific expression (*Zhang et al., 2018*). The predominant and equivalent male and female sturgeon gonads expression of putative piRNAs suggest the key role of piRNAs (rather than miRNAs) in sturgeon gonad development and sex differentiation by acting through piRNA-Piwi protein compound pathways.

To overcome the drawbacks of RNA-seq (Illumina sequencin), such as the potentiality of sequencing errors and be influenced by raw data processing before small RNA identification (*Git et al., 2010*), the expression of 1,871 putative piRNAs was validated by an independent

microarray platform. Data showed significantly sex-biased expression, including 124 that were specifically highly expressed in ovaries and 187 in testes (Fig. S2 and Table S5). There were no significant differences in the size of sex-biased sturgeon putative piRNAs. A previous study has shown that hsa-piR-020485 and hsa-piR-019825 (orthologs of ASY-piR-342 and ASY-piR-1706) are significantly associated with sex (Yuan et al., 2016), a result consistent with our observation of ASY-piR-342 and ASY-piR-1706 up-regulation in sturgeon ovaries (Table S5). Moreover, we found that 1,335 putative piRNAs were co-expressed in male and female sturgeon gonads (including 4 orthologs of known piRNAs), thus suggesting the crucial role of piRNAs in sturgeon physiological processes. The identification of sex-biased piRNAs combined with the co-expressed piRNAs in male and female gonads provides further insight into the molecular mechanisms of sturgeon gonadal development and sex differentiation.

Annotation of piRNA-generating genes suggests that putative piRNAs are involved in multiple reproductive processes (i.e., sexual reproduction, mating/reproductive behavior, and post-mating regulation of female receptivity). Moreover, 29 enriched KEGG pathways, including those relates to common challenges in aquaculture, such as pathogen infection, multiple diseases, and antigen processing, were identified (Table S3). Further enrichment analysis indicated that they probably participate in reproduction, mainly via sex-specific expression (Fig. 3 and Table S4).

## CONCLUSION

This study greatly increase the knowledge of small regulatory RNAs in the sturgeon *A. Schrenckii*. We have identified a large number of potential novel piRNAs and provided the first description of the presence of piRNAs with likely roles in sturgeon gonadal development and sex differentiation. Our data demonstrate that sturgeon putative piRNAs, similar to those of zebrafish, are predominantly expressed in both male and female sturgeon gonads, thus supporting a potentially conserved molecular function for piRNAs in sturgeon gametogenesis between sexes. Moreover, the sex-specific expression of putative piRNAs suggests that they function in sturgeon gonad development and sex differentiation. Furthermore, these gender-related piRNAs are the candidates to be developed as DNA bio-marker for early sex-determination with less damaging to sturgeon. Finally, information uncovered by this study aids in understanding sex determination in vertebrates.

### Funding
This work was supported by the Planning Funds of Scientific and Technological of the Guangdong Province (2016B070701016), the GDAS Special Project of Science and Technology Development (2019GDASYL-0104017), GDAS Special of Science and Technology Development (2018GDASCX-0107), and the National Natural Science Fund of China (31802279 and 31872499). The funders had no role in the study design, data collection and analysis, decision to publish, or preparation of the manuscript.

## Grant Disclosures

The following grant information was disclosed by the authors:

Planning Funds of Scientific and Technological of the Guangdong Province: 2016B070701016.

GDAS Special Project of Science and Technology Development: 2019GDASYL-0104017.

GDAS Special of Science and Technology Development: 2018GDASCX-0107.

National Natural Science Fund of China: 31802279, 31872499.

## Competing Interests

The authors declare there are no competing interests.

## Author Contributions

- Lihong Yuan conceived and designed the experiments, performed the experiments, analyzed the data, contributed reagents/materials/analysis tools, prepared figures and/or tables, authored or reviewed drafts of the paper, approved the final draft.
- Linmiao Li, Xiujuan Zhang and Haiying Jiang performed the experiments.
- Jinping Chen conceived and designed the experiments, contributed reagents/materials/-analysis tools.

## Animal Ethics

The following information was supplied relating to ethical approvals (i.e., approving body and any reference numbers):

The protocol was approved by the Committee on the Ethics of Animal Experiments of Guangdong Institute of Applied Biological Resources (GIABR2014008).

## Data Availability

All small RNA data series have been submitted to the SRA database under accession numbers SRR3180645, SRR3180649, SRR3180651, and SRR3180713. The microarray data have been deposited in the ArrayExpress, GEO database under accession number GSE83840.

## Supplemental Information

Supplemental information for this article can be found online at http://dx.doi.org/10.7717/peerj.6709#supplemental-information.

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
