# Peer review of "Identification and differential expression of piRNAs in the gonads of Amur sturgeon (Acipenser schrenckii)"

_PeerJ, doi:10.7717/peerj.6709_

## Round 0.1 · original submission · Major Revisions

Both reviewers found your manuscript interesting but raised several points that must be addressed before it is suitable for publication. There are several issues with the clarity and quality of the exposition that the reviews examine in detail.

Reviewer 2 raises an important point about the interpretation of the findings (see section "Validity of the findings"). I also agree that the significance of the results is overstated. Reviewer 2 suggests performing some experiments to support some of the conclusions. These experiments may be beyond the scope of this manuscript; if performing them is not possible, the manuscript should be extensively revised to reflect the speculative nature of the interpretation of the findings.

In addition to the points raised by the reviewers, it is important to point out the the short RNAs described in the study are putative piRNAs, since their identity relies on bioinformatic inference and is not supported by functional studies such as binding to Piwi proteins. Discussion of the findings should reflect this perspective.

Finally, the short RNAs described here have characteristics of prepachythene piRNAs, which, at least in mammals, are generally associated with retrotransposon sequences. This association is neither found nor its absence discussed in the study. Please address this point.

Reviewer 1 ·

Basic reporting

'no comment'

Experimental design

'no comment'

Validity of the findings

'no comment'

Additional comments

Overall, the manuscript is well written and easy to follow. The datasets and analyses are described sufficiently and the presented results are convincing. There are few places, as outlined below, that I recommend the authors to review.

* In the abstract,:"sturgon" should read "sturgeon" (Line 33, 36 and 39).

* In the introduction, the authors should mention what is known about piRNA generation in sturgeons. Molecular mechanisms, like ping-pong model (as mentioned in Line 318) should be mentioned here as well.

* The section 'Small RNA annotation and piRNA identification' is not clear. It says, previously published transcriptome was used to map small RNAs. Does "clean unique reads" refer to small RNAs sequenced for this study? In that case, why were the redundant reads removed from small RNA libraries? I can see why one would remove redundant reads from mRNA-seq, but given this is small RNA-seq wouldn't one expect to see many small RNA reads of exact same length and nucleotide composition to be sequenced? Please make it more clear in the paragraph every time you refer to small RNA seq datasets generated for this study.

* In that same paragraph, the steps taken to remove certain small RNA reads should be stated instead of just saying 'as previously described' (Line 157).

* The link to the piRNA database provided in line 165 is not working.

* In line 211, the authors should write clearly what transcriptome reference sequences were used.

* Re-write the Lines 210-211: "After these results were mapped to the A. schrenckii transcriptome reference sequences, 3.1 - 3.4 × 10^6 perfectly matched small RNA reads remained", with e.g.,
'About 3.1 - 3.4 × 10^6 small RNA reads mapped to the A. schrenckii transcriptome reference sequences with a perfect match.'

* Re-write the Line 220: "A total of 5.7 - 6.2 × 10^5 piRNAs reads, corresponding to 16.6 - 19.8% perfectly matched small RNA reads, were obtained." e.g.,
'In total, we identified 5.7 - 6.2 × 10^5 piRNAs reads in our sequencing results, that account for 16.6 - 19.8% of perfectly matched small RNA reads.'
To further improve the clarity of that sentence, it may be worth repeating, what was the criteria to claim that a given read is indeed a piRNA. Give a short explanation and some references, too.

* Line 221: The authors identify 875,679 putative novel piRNAs from the testis and ovary libraries. Give a short explanation why these reads are considered 'putative novel piRNAs'. What fraction of the reads these novel piRNAs account for? Also, refer the reader back to the methods, by writing, e.g., 'novel piRNAs were predicted by following steps described in the methods section'.

* In Figure 1A.: You should state clearly, whether the figure considers only known piRNAs, only putative novel piRNAs or all together.

* Line 225: It is unclear how the "rate of sex-specific expressed piRNAs was up to 87%" was calculated. There should also be some description of what it implies.

* Figure 2A: Is this figure showing just '875,679 putative novel piRNAs'? Line 236 would suggest so, but the figure legend says it is all small RNAs.

* Line 236: "26-30 nt in length according to the potential piRNA reads" >>> it may be better to write it as follows:
>>> "Both male and female sturgeon gonads displayed a peak in reads of 26-30 nt in length. This is an expected length distribution for potential novel piRNA reads."

* Line 239-240: "showed a peak in reads of 20-24 nt in length corresponding to miRNA reads" >>> ...showed a peak in reads of 20-24 nt in length which is an expected length distribution for miRNA reads.

* Line 241-242 "Moreover, both ovary and testis samples had a strong 5’ U bias in reads of 20-24 nt (miRNAs) and 26-30 nt (piRNAs) in length, whereas only one peak of 5’ U was found in reads of 20-24 nt among somatic samples (Figure 2B-2D)." >>> I think the authors meant "... whereas only reads of 20-24 nt among somatic samples had a 5’ U bias."
Following up, does this figure only include known/infered miRNAs and piRNAs or is it all reads of that size and simply called miRNAs and piRNAs? Please make it clear in the text.
The legend to the Figure 2 should also be more precise and clear about what is being plotted.

* Line 248: "After mapping onto the A. schrenckii transcriptome, " >>> What exactly was mapped? Just the 875,679 putative novel piRNAs? This should be clearly stated.

* Line 248: What criteria/threshold was applied to call a gene a piRNA-generating gene? Also, give a reference to the list of genes used in the mapping.

* Line 250: "up to 4,181 unique piRNAs reads per gene (mean of 18 piRNAs per gene)." The values provided should be scaled at least to the library size, rather than just giving raw number of reads.

* Line 254: What tool was used to perform GO analysis?

* Line 260: Do the authors mean "piRNAs" or sex-specific piRNA-generating genes? Double check.

* Line 263: Name the statistical test used in the calculation of the p-value. Given that the p-value is just 0.01 it is probably better to just say that a statistical difference was detected, although not to a very high degree.

* Line 266: How the 1871 piRNAs were chosen? Repeat briefly and point back to the methods.

* Line 273: "ASY-piR-122 with median of normalized intensity 13.33 in ovaries vs. 4.91 in ovaries" -- I believe the second value refers to testes.

* Line 275: "Further analysis indicated no significant difference in piRNA size distribution among sex-biased piRNAs " --> A paragraph before the authors write there is a difference is sex-specific piRNAs which creates a confusion at first. It would make it easier if the authors could stress more the difference between sex-specific and sex-biased piRNAs.

* Line 278-280: "The correlation analysis indicated that the expression of piRNAs detected by Illumina sequencing was correlated with those identified by microarray (for 1,871 piRNAs: r = 0.202, P = 0.000; 311 sex-biased piRNAs: r = 0.489, P = 0.000)."
>>> Maybe it is better to highlight the fact that sex-biased piRNAs were more correlated. Are these also more highly expressed in the Illumina dataset?

* Line 318: It is unclear how Figure 1A points to a Ping-Pong model.

* Line 319: "we found the presence of piRNAs in both male and female sturgeon gonads, but not in somatic tissues (Figure 2A)". Figure 2A only shows the size distribution of the sequenced reads. This sentence should probably read:
>>> "Small RNA reads with characteristics typical to piRNAs were found in both male and female sturgeon gonads, but not in somatic tissues."

* Again, the statement in Line 323 is based on p-value 0.01: "significant differences in the sizes of sturgeon sex-specific piRNAs (read counts ≥ 10) were observed."

* The statement in Line 345: again, I think it would benefit the readers to write clearly what is the difference between sex-biased and sex-specific piRNAs.

* Figure 4: The authors should include some information regarding the clustering method used and what metric was used for calculating the distance. This can either be in the legend, or somewhere in the main text.

Other comments:

* Increase the label size in the supplemental Figure S1.
* Table S1: enlarge the column 'clean reads' so that the second line can be read.
* Increase the resolution of Figure 1.
* Font size in Figure 2 could be bigger.
* Figure 3 could be split in two or three parts and plotted along side. This way one could make the text bigger and thus more readable.
* Figure 4: The heatmap, in its current state, does not seem to give any more information than said. If there are clusters that are particularly interesting, it may be worth highlighting those and give some description. One could also try including expression of these piRNAs as determined by Illumina sequencing for comparison.

Reviewer 2 ·

Basic reporting

English is clear enough throughout the manuscript

The authors used small RNA sequencing and microrrays to identify sturgeon piRNAs by small RNA sequencing in Illumina HiSeq and validate their expression with microarray.

Knowledge related to sexual development in sturgeon may have a high economic impact in farming profit. Thus, I agree that it is worth to study sexual development in sturgeons.

In the introduction (lines 83 to 89) they emphasize the difficulties derived from the instability of sex development, the lack of knowledge regarding sex chromosomes and the difficulty of molecular studies. I do not see why all this facts support that sturgeons are good models for studying the mechanism of sex determination in vertebrates. Model organisms should facilitate work. I suggest removing the sentence in lines 85-86 suggesting that they are a good model.

At the end of the introduction (lines 90 and above) they introduce the role of piRNAs in germline development and transposon silencing but the most recent citation in this part is form 2015. There are more recent reviews to cite such as doi: 10.1242/dev.161786 or doi: 10.1530/REP-18-0218 for more different possible roles of piRNAs.

Figures are not of sufficient quality. They mix different fonts. For instance Fig 1A has different fonts in X and Y axes and Y axe label is misspelled, Fig 1B also mix different fonts, some of the number are blurry (as 58% and 30% in 2C) and other labels are difficult to read. Bar graph in Fig. 3 is too wide and GO terms can be hard to read. Maybe a shorter version with the most relevant term could be easiest to read and the complete list of GO Terms can be provided as supplementary material, arranged as a table with terms and number of genes instead of a bar graph. Fig 4 is also unreadable.

Experimental design

The experimental design is correct and the research is within the scope of the journal. The authors used small RNA sequencing and microrrays to identify sturgeon piRNAs by small RNA sequencing in Illumina HiSeq and validate their expression with microarray. All the methods are not fully described but they cite other papers for more information related to the methodology, providing sufficient information to replicate. The main contribution of this paper is to provide bulk data of putative piRNAs in sturgeon.

Validity of the findings

The RNAseq piRNAs data provided are novel and may contribute to more research in the future. However, the further analyses presented in this paper are not really relevant. They show that some piRNAs have sex-specific expression and they suppose this is as a proof that sturgeon sex development is regulated by sex specific piRNA expression, but there is no support for such a statement as correlation do not means causality. They also perform a GO terms enrichment analysis showing enrichment in terms related to ovary or testis function and development. Although this may give some clues about what piRNAs may deserve further analysis, it does not demonstrate that they are really involved in such processes. The paper is not mature enough to be published. The GO and KEGG additional analysis of the data may provide clues regarding what piRNAs may have a real biological role and additional experiments should be done to corroborate that in fact they have. If such a proof is found it would demonstrate that these RNAseq data are really useful and that the analyses performed are able to identify biologically relevant piRNAs. I understand that performing these type of studies in sturgeon may be challenging. This is another reason of why they are not a good organism model.

Additional comments

The RNAseq data are novel and deserve publication. I agree that they may be useful in discovering relevant biological roles of particular piRNAs in some of the processes mentioned in the paper. I suggest that the authors use the preliminary results of their analyses to identify at least one example of biologically relevant piRNA and demonstrate its role, otherwise all the statements regarding the putative role of the identified piRNAs in relevant biological processes are merely speculative.

---

## Round 0.2 · Minor Revisions

Both reviewers found that the article was greatly improved, but that clarity could be further improved.

Reviewer 1 ·

Basic reporting

'no comment'

Experimental design

'no comment'

Validity of the findings

'no comment'

Additional comments

The paper has much improved from the previous version. The methods now contain more details necessary to be able to reproduce the results. The authors have also made a considerable effort to highlight the speculative nature of the analyzed piRNAs.

Below, I list some typos in order of appearance that could be corrected:

line 38> analyze not "analysis"> "...(GO) database and KEGG pathway database were searched subsequently to analysis the potential bio-function of sturgeon piRNAs"

line 91> "while" is misued. You can just delete it.

line 130> "three males and 3 females" > just use one notation type, either as a word or using numeric characters.

line 153> "successively" should be "successive"
"being" in 1), 2) and 3) is unnecessary

line 159> "previously" should read "previous"

line 248> "Comparing the total read counts of piRNA with that of miRNA expressed in sturgeon testis and ovary samples, we obtained a similar ratio of 4.7:1~5.5:1. Moreover, both piRNA (1:1) and miRNA (1.2:1) showed an equal ratio in testes vs. ovaries (Table 1)."

The authors should give an interpretation to this observation as well.
e.g. "This clearly indicates putative piRNAs are far more abundant class of small RNAs in both testis and ovaries compared to miRNAs."

While it is OK to compare read counts within the same sample for two different small RNA classes (since library size cancels out), it is not OK to compare read counts of the same class between two different samples. Each sample has a different library size that should be taken into account when making such comparisons. I suggest either correcting the "piRNA (1:1) and miRNA (1.2:1)" ratios or removing this sentence from the manuscript.

line 254> The authors could provide a total number of currently known genes in this specie for a context.

line 271> "Moreover, a statistical difference was detected in size distribution of sex-specific piRNAs (average 27.26 nt in testes vs. 27.78 nt in ovaries, t test)."
The authors fail to report the p-value. In any case, even if you see the "statistically significant" difference in length, it s unclear what it would mean to be 0.5 nt different. I suggest removing this statement from the manuscript.

line 281> the notation "|logFC| of 1~8.5" looks odd. Better to write it as: "1 < |logFC| < ~8.5". Similarly, "|logFC| of 1~4.1" can be written as "1 < |logFC| < ~4.1".

Reviewer 2 ·

Basic reporting

The paper and figures have been improved but some paragraphs still need some rewording.

“Putative piRNA” are redundantly repeated all over the manuscript. The sentences should be reworded to avoid this repetition. Suggestions for possible rewriting of some text follows:

Line 41..: A total of 875,679 putative sturgeon piRNAs were obtained, including 93 homologous to known piRNAs and hundreds showing sex-specific and sex-biased expression.

Line 43...: Further analysis showed that they are predominant in both the ovaries and testes and those with a sex-specific expression patern are nearly equaly distributed betwen sexes. This may imply a relevant role in stungeon gonadal development.

Line 48…: KEGG pathway and GO annotation analyses indicated that they may be related to sturgeon reproductive processes.

Once it is stated in line 41 that they are “putative piRNAs” ther is no need to repeat many more times in the same paragraph.


Line 191…: To validate the expression of putative piRNAs identified by Illumina sequencing, we selected: 1) 1,092 that exhibited significantly different expression in ovaries vs. testes with │logFC│≥1.5, P ≤ 0.01, Padj ≤ 0.05, and read counts > 10; and 2) 779 highly expressed with read counts 50 ~ 4,646 in ovaries or 50 ~ 8177 in testes. For each of the six RNA samples (3 ovaries and 3 testes, extracted above) assessed, 4 µg of total RNA were used to hybridize with the microarray chip.

Line 250…: Further analysis indicated that their rate of sex-specific expression (only expressed in ovary or testis) was up to 87%

Line 252…: and the ratio of ovary- vs. testis-specific putative piRNAs was nearly 1:1 (43.99%, 385,222/875,679)

Line 254…: Moreover, we found a simmilar percentage of sense and antisense strand putative piRNAs (42.99%, 376,487/875,679 vs. 57.01%, 499,192/875,679, respectively)

Line 305…: A total of 311 showed significantly biased expression between sexes

Line 308…: 124 where specifically up-regulated..

Line 313…: In contrast, the most expressed n the ovaries was ASY-piR-122…

Line 315…: Moreover, 1,335 were co-expressed in both the testes and ovaries…

Line 316…: Further analysis indicated no significant difference in size distribution among sex-biased putative piRNAs…

Line 318..: The Illumina sequencing and microarray expression of putative piRNAs was especially correlated for 311 sex-biased piRNAs (r = 0.489, P = 0.000).

Line 331…: and hundreds of them showed sex-specific (only expressed in sturgeon testis or ovary) and sex-biased expression, validated by microarray.

Line 352..: A total of 875,679 sturgeon putative piRNAs were identified, including 93 homologous to known piRNAs.

Line 358…: Approximately 86% of reads were sequenced only once,…

Line 368…: We show that both expression and distribution on gene regions of putative piRNAs are equally distributed between testis and ovary (Figure 1B and 1C).

Line 370…: Significant differences in the sizes of sturgeon sex-specific putative piRNAs (read counts ≥ 10) were observed (t test, P <0.01), as also demonstrated in human sex-specific piRNAs, which show gender differences in size distribution and thus affect binding with Piwi proteins (Williams et al., 2015).

Line 374…: Furthermore, the prevailing expression of putative piRNAs in gonads has also been found in many other organisms (……) and provides insight into …….
Line 386…: The predominant and equivalent male and female sturgeon gonads expression of putative piRNAs suggest……

Line 394…: Data showed significantly sex-biased expression, including 124 that were specifically highly expressed in ovaries and 187 in testes (Figure S2 and Table S5).

Line 412..: Further enrichment analysis indicated that they probably participate in reproduction, mainly via sex-specific expression (Figure 3 and Table S4).

Experimental design

As in the previous version the experimental design is correct

Validity of the findings

It has been made clear that the piRNAs are only putative because no evidence of biological significance has been provided.

---

## Round 0.3 · accepted · Accept

Your responses to the latest reviews satisfy all the reviewers' requests and your paper is now suitable for publication.

#